# Evaluation of a Deep Learning Approach to Differentiate Bowen’s Disease and Seborrheic Keratosis

**DOI:** 10.3390/cancers14143518

**Published:** 2022-07-20

**Authors:** Philipp Jansen, Daniel Otero Baguer, Nicole Duschner, Jean Le’Clerc Arrastia, Maximilian Schmidt, Bettina Wiepjes, Dirk Schadendorf, Eva Hadaschik, Peter Maass, Jörg Schaller, Klaus Georg Griewank

**Affiliations:** 1Department of Dermatology, University Hospital Essen, 45147 Essen, Germany; philipp.jansen@ukbonn.de (P.J.); dirk.schadendorf@uk-essen.de (D.S.); eva.hadaschik@uk-essen.de (E.H.); 2Department of Dermatology, University Hospital Bonn, 53127 Bonn, Germany; 3Center for Industrial Mathematics (ZeTeM), University of Bremen, 28359 Bremen, Germany; otero@uni-bremen.de (D.O.B.); leclerc@math.uni-bremen.de (J.L.A.); maximilian.schmidt@uni-bremen.de (M.S.); pmaass@uni-bremen.de (P.M.); 4Dermatopathologie Duisburg Essen GmbH, 45329 Essen, Germany; nicole.duschner@gmail.com (N.D.); b.wiepjes@gmx.de (B.W.); dermatohistologie@googlemail.com (J.S.); 5Dermatopathologie bei Mainz, 55268 Nieder-Olm, Germany

**Keywords:** Bowen’s disease, seborrheic keratosis, artificial intelligence, U-Net, digital pathology, whole-slide image (WSI), computer-aided diagnosis (CAD)

## Abstract

**Simple Summary:**

Bowen’s disease (malignant) and seborrheic keratosis (benign) are frequent cutaneous neoplasms. Our study assessed the potential of artificial intelligence to distinguish these entities histologically. A dermatopathologist trained deep learning network diagnosed Bowen’s disease and seborrheic keratosis with AUCs of 0.9858 and 0.9764 and sensitivities of 0.9511 and 0.9394, respectively. The algorithm proved robust to slides prepared in three different labs and two different scanner models. Nevertheless, challenges, such as distinguishing irritated seborrheic keratosis from Bowen’s disease remained. We believe our findings demonstrate that deep learning algorithms can aid in clinical routine; however, results should be confirmed by qualified histopathologists.

**Abstract:**

Background: Some of the most common cutaneous neoplasms are Bowen’s disease and seborrheic keratosis, a malignant and a benign proliferation, respectively. These entities represent a significant fraction of a dermatopathologists’ workload, and in some cases, histological differentiation may be challenging. The potential of deep learning networks to distinguish these diseases is assessed. Methods: In total, 1935 whole-slide images from three institutions were scanned on two different slide scanners. A U-Net-based segmentation deep learning algorithm was trained on data from one of the centers to differentiate Bowen’s disease, seborrheic keratosis, and normal tissue, learning from annotations performed by dermatopathologists. Optimal thresholds for the class distinction of diagnoses were extracted and assessed on a test set with data from all three institutions. Results: We aimed to diagnose Bowen’s diseases with the highest sensitivity. A good performance was observed across all three centers, underlining the model’s robustness. In one of the centers, the distinction between Bowen’s disease and all other diagnoses was achieved with an AUC of 0.9858 and a sensitivity of 0.9511. Seborrheic keratosis was detected with an AUC of 0.9764 and a sensitivity of 0.9394. Nevertheless, distinguishing irritated seborrheic keratosis from Bowen’s disease remained challenging. Conclusions: Bowen’s disease and seborrheic keratosis could be correctly identified by the evaluated deep learning model on test sets from three different centers, two of which were not involved in training, and AUC scores > 0.97 were obtained. The method proved robust to changes in the staining solution and scanner model. We believe this demonstrates that deep learning algorithms can aid in clinical routine; however, the results should be confirmed by qualified histopathologists.

## 1. Introduction

Seborrheic keratosis (SK) is among the most common benign epithelial lesions affecting the majority of patients older than 60 years [1,2]. SK may present as papules, macules, or plaques and vary in color from tan to black with a verrucous, “stuck-on” appearance. Additionally, SK may show exophytic or hyperplastic growth [2,3,4]. Due to their high variability in clinical and dermatoscopic presentation, SKs are frequently excised to exclude malignant cutaneous tumors [5]. Bowen’s disease (BD) is a cutaneous squamous cell carcinoma in situ and one of the most frequent malignant tumors worldwide. The clinical appearance ranges from slow-growing, persistent reddish-brown patches to slightly raised plaques of dry skin. Based on unique histopathological characteristics regarding the absence or presence of crowded nuclei, suprabasal mitoses, necrotic keratinocytes, or parakeratosis housing plump nuclei, the majority of SK and BD can be differentiated easily by histopathological analysis. However, due to the broad spectrum of morphological manifestations in both entities, histopathological differentiation may be challenging in a number of cases. Factors such as leukocytic infiltration or morphological alteration of tumor cells as a result of inflammation can make a histological distinction between these entities difficult. Additionally, benign clonal cell proliferations can be present in SK, resembling BD. Aside from histological similarities, small biopsies with limited amounts of tissue being available for analysis can further impede the reliable histological differentiation of SK and BD [6,7,8]. Correctly distinguishing these lesions is, however, essential, as therapeutical approaches differ. Complete treatment of malignant tumors such as BD is mandatory as those will continue to grow and may evolve into invasive neoplasms with significant morbidity, potentially even mortality. In contrast, the treatment of SK is primarily for aesthetic purposes [6]. An inappropriate diagnosis may thus result in overtreatment or undertreatment, both having potentially serious long-term unnecessary effects for affected patients.

Despite increasing numbers of malignant and benign tumors, the number of dermatopathologists is decreasing. In recent years, artificial intelligence (AI) has been implemented in different disciplines of medicine. AI appears to have the potential to improve and accelerate diagnostic work-flows and may thus defuse the problem of a continuous shortage of dermatopathologists. Deep learning algorithms and convolutional neural networks (CNNs) for object detection and segmentation have rapidly been utilized in dermatology for the classification of dermoscopic images [9,10], to differentiate various benign and malignant cutaneous lesions.

The digitization of entire histology glass slides to obtain whole-slide images (WSIs) is now being adopted across the world in pathology labs. This has also enabled the development of AI methods for histopathology. One of the main challenges is the size of the WSI (approximately 40,000 px × 20,000 px at a 20× level of magnification) in comparison to dermoscopic images (sizes vary from 300 px×300 px up to 1000 px×1000 px). Another challenge is the variation of slide staining protocols across laboratories, which might affect the robustness of the models.

In the present study, we implement and evaluate a U-Net-based [11] segmentation algorithm to differentiate SK from BD on WSIs obtained from three different centers. Our algorithm produces a detailed segmentation map of the WSIs, where the pathologists can directly see which parts of the tissue (at the pixel level) are identified as one diagnosis or another. The algorithm is meant to support the dermatopathologists in their daily routine by providing an initial diagnostic suggestion and additional information.

### Related Methods

In histopathology, networks based on the U-Net architecture have gained popularity. Numerous working groups obtained impressive results on the CAMELYON challenge [12] for detecting breast cancer metastasis in lymph nodes by using such architectures [13,14]. Li and Lu [15] proposed a model for rapid detection and refined segmentation of breast cancer regions applying a ResNet101 [16] or MobileNetV2 [17] base, followed by a U-Net architecture to refine the segmentation. Another architecture called HookNet [18] combines context and details via multiple branches of encoder–decoder convolutional neural networks. Other, simpler architectures have also obtained promising results [19,20].

In dermatopathology, deep learning approaches have been applied for the segmentation of both melanocytic [21] and non-melanoma lesions, including basal cell carcinoma (BCC) [22]. A U-Net architecture automatically segmenting the epidermis in histological images has also been implemented [23]. Olsen et al. [20] applied a deep-learning-based segmentation method based on the VGG architecture to differentiate three common diagnoses: nodular BCC, SK, and dermal nevus. Ianni et al. [24] trained a model differentiating four groups of lesions: melanocytic, basaloid, squamous, and “other”. Evaluating the model with data from three different laboratories not used for training, a 78% accuracy was achieved. A two-stage AI-based system for the automatic multi-class grading of oral squamous cell carcinoma and for the segmentation of the epithelial and stromal tissue has also been presented [25].

Despite those numerous studies, significant challenges remain to transfer AI from research into clinical practice [26]. In this work, we present a deep learning solution to differentiate some of the most common diagnoses in dermatopathology: BD and SK. To the best of our knowledge, no work has yet investigated the automatic identification and segmentation of BD, nor its differentiation from SK. These are two of the most common diagnoses seen in the daily clinical routine, and their distinction can on occasion be challenging for dermatopathologists. They are both intra-epidermal (in situ) lesions that can exhibit similar patterns. Our system can assess the WSI and report if it detects any BD or SK lesion at all. This can be very useful to assist the dermatopathologist in the assessment of resection margins. Other works in the literature [20,24,25] mainly focus on analyzing WSIs that contain some lesion, and the task is to classify it into 1–4 classes.

## 2. Materials and Methods

### 2.1. Ethics Approval

The study was performed in accordance with the approval of the ethics committee of the University of Duisburg-Essen (IRB-Number 20–9196-BO).

### 2.2. Dataset Creation

SK and BD included in the study were samples from the MVZ Dermatopathologie Duisburg Essen GmbH, Essen (Center 0), University Hospital Essen, Department of Dermatology (Center 1), and Dermatopathologie bei Mainz (Center 2). Samples were excised for histopathological analysis during daily routine clinical work, and slides with tissue were processed according to standard staining protocols. The cases represented a mix of biopsies, partial tumor excisions, and complete excisions. Biopsies were either spindle-excision, punch, or curettage. Based on the diagnosis given during the clinical routine, the diagnosis of SK or BD was reevaluated by at least one board-certified dermatopathologist. Training slides were selected for cases with unambiguous morphology and diagnosis. Slides were digitized by applying either a Leica^®^ Aperio AT2 or Hamamatsu Nanozoomer S360 slide scanner; see Table 1.

Annotation of the WSI was performed in an efficient semi-automatic process between pathologists and the AI using our experimental platform DigiPath-Viewer, which is being developed at the Center for Industrial Mathematics in Bremen. Based on a few entirely manually annotated samples, the AI model was trained to make initial suggestions for segmentation. Annotation suggestions by the AI model followed, requiring only minor adjustments by the pathologists.

### 2.3. Special Test Dataset from Center 0

All 865 WSIs belonging to the test dataset from Center 0 correspond to all cases digitized in that laboratory between 1 January and 8 March 2022. All cases were used without pre-selection and were automatically pre-diagnosed with a previous version of the model integrated into the DigiPath-Viewer. Diagnoses were checked and corrected if necessary by the dermatopathologists parallel to the daily routine work. To date, this platform is only used for research purposes.

The SK cases from Center 0 were also classified as irritated or not irritated SK, enabling us to compute metrics on those subsets. Both the training and test sets from Center 0 also include WSIs having sections without any disease, identified in the manuscript as “Normal”. The exact distribution of WSIs and sections of Center 0 is depicted in Table 2.

### 2.4. Architecture and Training of the Model

The implemented method works end-to-end, i.e., it receives a WSI as the input and delivers the whole segmentation map. The first step is to differentiate tissue from the background by a combination of classic image filters, morphological operations, and a threshold method to produce a binary mask. Due to the immense size of the WSI, it is not possible to feed the entire data at once as the input to a neural network due to the limited memory on the graphics card (NVIDIA RTX A6000 with 48 GB). Therefore, it is common practice [14] to extract tiles, feed them independently to the network, and combine the results afterward. In Figure 1, a WSI with BD from Center 0 is shown, where one of the tissue sections was tiled with our method. We used tiles of size 1024 μm × 1024 μm at a 10× level of magnification with at least 300 μm overlap in each direction. At 10×, it holds that 1 μm=1 px, so the size of the tiles in pixels is 1024 px × 1024 px. Figure 2 shows some of the patches extracted from the tiled section in Figure 1, whereas Figure 3 shows patches extracted from a WSI from Center 2. In the latter, one can observe some differences in the style of the images due to stain variations and that the glass slides were digitized with scanners from different manufacturers.

The neural network architecture is a simplification of the original U-Net [11]. The model is made of an encoder, which in our case is a ResNet34 [16], and a decoder that only has two levels of up-sampling; see Figure 4. In the encoder, the spatial dimensions go down from 1024 px × 1024 px to 32 px × 32 px. The output of the network contains two probability heat maps, one for each diagnosis, BD and SK, for size 128 px × 128 px, i.e., each pixel in the output corresponds to a region of 8 px × 8 px in the original tile. The reason for doing this, instead of utilizing an output the same size as the original input image, is to reduce the complexity of the model and the number of trainable parameters. The immediate gain is that the model requires less memory to run and enables training with larger tile sizes. Only half of the input size could have been trained applying the original U-Net architecture. Tile size is critical as only a single image tile can be evaluated by the network to make a diagnosis. To correctly diagnose BD or SK, it is crucial to analyze as much tissue as possible.

Network training was performed in a fully supervised setting. A large dataset of tiles and their corresponding pixelwise labels (masks) were created and split into training (80%) and validation (20%) parts. The split was performed such that all tiles from the same WSI were either in one part or the other. The model was trained using the Adam [27] optimization method for focal loss minimization [28] (γ=2) with an initial learning rate of $10^{-4}$. Additionally, data augmentations were performed, including random rotations, elastic deformations, noise, blur, brightness, contrast, and color variations to increase the variety in the training set and the model’s robustness.

We used the PyTorch framework [29] and a distributed parallel setting with 4× NVIDIA RTX A6000 and an effective batch size of 256. During training, the intersection over union (IoU) score was monitored, both on the training and validation dataset; see Figure 5. Optimization converged after 40 epochs without having observed over-fitting.

### 2.5. Hyper-Parameter Selection

For each call “BD vs. all” or “SK vs. Normal” on the section level, i.e., after combining the results from the single tiles, two hyper-parameters were used: the threshold for making the probability heat map and the minimum positive area to make the call (cf. Figure 6). A line search over a range of meaningful values was performed using the training dataset from Center 0 to estimate the optimal minimal area. For each candidate value, the receiver operating characteristic (ROC) curve of the model and the area under the curve (AUC) were computed. The area yielding the highest AUC score was selected. AUC scores measure how well the model performs in binary decision tasks, with the advantage of being threshold independent, i.e., a threshold does not need to be selected beforehand. Figure 7 shows the results of the scores for each of the candidate areas in the grid. The ROC curve is obtained by computing for every threshold the true positive rate (TPR), which is equivalent to the sensitivity of the model, and the false positive rate (FPR), which is equivalent to 1−Specificity. They are defined as
$$\mathrm{TPR}=\frac{\mathrm{TP}}{\mathrm{TP}+\mathrm{FN}}\;,$$
$$\mathrm{FPR}=\frac{\mathrm{FP}}{\mathrm{FP}+\mathrm{TN}}\;,$$
where TP= true positives, FP= false positives, TN= true negatives, and FN= false negatives.

Once the minimum positive area hyper-parameter was chosen, the threshold to make the cut in the probability heat map was selected by choosing the threshold yielding the maximum $F_{\beta }$ score (cf. Figure 8). It is a weighted harmonic mean of the model’s precision and recall/sensitivity. The parameter β is chosen to make the sensitivity β-times as important as precision. In this case, we chose β=2 for BD to attribute higher importance to the sensitivity and the standard β=1 for SK.
(1)$$F_{\beta }=\left(1+\beta^{2}\right)\frac{\mathrm{Recall}\;\cdot \;\mathrm{Precision}}{\mathrm{Recall}+\beta^{2}\;\cdot \;\mathrm{Precision}}.$$

The selected areas and their corresponding AUC scores, as well as the selected thresholds and the $F_{1}$ scores are depicted in Table 3. In Figure 9, a selection of tiles is shown with their corresponding model BD output. Before combining the results, post-processing with morphological filters to the probability maps was applied to make them smoother and remove minor artifacts (third column). For reference, we also include the result after applying the selected threshold (fourth column).

## 3. Results

### 3.1. BD vs. SK

The main goal of this study is to investigate to what extent our deep learning approach can differentiate BD from SK. In order to obtain statistics on how well the model performs in this task, all normal tissue sections without any of those labels were filtered out, and the model was used to predict whether there is BD in the section. This is what is identified as “BD vs. SK”. Three test datasets, each from one of the laboratories, were used to evaluate the model. The results are depicted in Table 4 and Figure 10 and Figure 11.

The results show that the model can make the distinction between BD and SK with high confidence, especially on the data from Centers 1 and 2, which were not involved in the training process of the deep learning model. This performance transfer to other labs highlights the generalization capability and robustness of the approach. An example segmentation for a BD test cut from Center 0 is shown in Figure 12.

### 3.2. BD vs. All

This section evaluates how well the model distinguishes BD from all other tissue types included in the study: normal tissue, SK, only non-irritated SK, and only irritated SK. For this evaluation, we only used the test set from Center 0. The corresponding ROC curves are provided in Figure 13, and the metrics are depicted in Table 5.

Table 5 shows that the model has a comparatively low specificity for distinguishing between BD and irritated SK. In the following, we take a closer look at this phenomenon and propose an adaption of the model.

#### Relative Area Approach

We observed that the model makes mistakes by predicting irritated SK as BD. However, in most of those cases, the model recognizes only a small part as BD and the rest as SK. Therefore, we implemented another rule for differentiating BD from SK based on the relative area *r* recognized as BD:(2)$$r=\frac{100\;\cdot \;\mathrm{area}(\mathrm{BD})}{\mathrm{area}(\mathrm{BD})+\mathrm{area}(\mathrm{SK})}\;.$$

We then changed the condition for the BD call to be
(3)$$\mathrm{area}\left(\mathrm{BD}\right)\geq \delta_{1}\;\mathrm{and}\;r\geq 20\;.$$

Table 6 reports the results. The relative area approach helps to improve the specificity of the model, as Figure 14 shows. Especially for the differentiation of irritated SK, a value above 0.8 is now achieved. This comes at the cost of a slightly decreased sensitivity compared to the previous results in Table 4 and Table 5. Additional examples of the model’s prediction, including the relative BD area, are shown in Figure 15 and Figure 16.

### 3.3. SK vs. Normal

We also evaluated how well the model differentiated SK from normal tissue. For this, we only used the test set from Center 0, which also contains many normal tissue sections and excludes sections containing BD. The results can be found in Figure 17 and Table 7. The distinction between SK and normal tissue works reliably. A segmentation example of an SK section is shown in Figure 18.

## 4. Discussion

SK and BD are among the most common lesions in older patients (>60). Due to the high variability in clinical appearance, these tumors are frequently biopsied or excised. Differentiating benign SK from malignant BD, which can progress to invasive squamous cell carcinoma (SCC) [30], is crucial. The majority of histological samples can easily be distinguished on the basis of hematoxylin and eosin staining (H&E), requiring no additional stainings. However, the infiltration of leukocytes as a sign of irritation or inflammation and clonal cell proliferation in SK can occasionally make correctly differentiating those two tumors challenging. Additionally, cases of BD developing in SK have been described [31,32]. It remains controversial whether this is a causative or collision phenomenon. Sufficient tissue must be available to identify the correct diagnosis. Thus, little tissue as a consequence of small biopsies may additionally be challenging even if several serial sections or immunohistological stainings are performed. In complex cases, neither histological parameters, nor a single immunohistochemical marker are sufficient to distinguish these two entities. Some have suggested that a reliable distinction can be made with a panel of immunohistological markers [33]. However, this is both cost-intensive and time-consuming. Especially when little tissue is available and the diagnosis may depend on the morphological traits of a few cells, artificial intelligence (AI) might be helpful to draw the dermatopathologists’ attention to those areas of interest.

In most cases, our established algorithm could classify the tumor (BD vs. SK vs. normal tissue) in accordance with the diagnoses made by the dermatopathologists. One of the great fears of a pathologist is missing a diagnosis. A pathologist recognizing things as hard to classify, means one knows there is a problem and can search for a solution. It is more dangerous to not realize a problematic case and miss it. This is precisely the scenario where we believe AI has its potentially greatest use in the foreseeable future. An example is the previously mentioned collision tumors, where a small portion of the slide has a malignant neoplasm. The AI recognizing it, and suggesting review by the pathologist could ensure correct diagnosis. Our results show that the AI was able to detect BD (BD vs. all) with an AUC of 0.9858 and sensitivity 0.9511 on a large test set from Center 0 (3213 tissue sections). This means the AI is not only able to differentiate BD from SK, but also to decide if there is any BD lesion at all. This could be useful to assist dermatopathologists in the assessment of resection margins.

Our findings demonstrate that the deep learning model can distinguish between BD and SK with an AUC of 0.9774 on an uncurated test dataset (Center 0) and an AUC of 0.9988 and 0.9941 on a selection of cases from two other institutions with data samples that were not involved in the training process (Center 1 and Center 2). Still, the specificity of the algorithm to distinguish BD from irritated SK—with the relative area approach—was 0.81 in contrast to a specificity of >0.90 for the differentiation of BD vs. SK (not irritated) vs. normal (cf. Table 5). We conclude that the low number of irritated SK included in the study is one reason for the low sensitivity. The algorithm could not be sufficiently trained to recognize these lesions. Thus, enlarging the training cohort of irritated SK would likely improve the algorithm and enhance the sensitivity.

Another interesting aspect is collision tumors, where both BD and SK are present. Recognizing the malignant disease (BD) will be more critical for the patient, so we prioritized BD detection as the initial step in our algorithm. Incorrect classification of irritated SK areas as BD led to incorrect classification of the entire tumor. As mentioned above, this caveat should be largely minimized by additional training. However, actual collision tumors exist, and these cases should not be classified as solely one diagnosis. This is something we will work on further. However, we believe this represents one of the benefits AI can offer as an aid for routine pathologists. If, for example, 90% of the presented lesion is SK, but there is also BD present in small amounts, it is conceivable the dermatopathologist might miss the, for the patient, actually relevant BD diagnosis and just diagnose SK. A red flag by the AI showing the relevant area could help ensure the pathologist does not oversee any relevant areas.

## 5. Integration in Daily Routine

One can generally wonder how best to implement the algorithm in daily routine. In conventional current routine work, many pathologists first look at the histology and, only after gaining an initial impression based solely on the morphology, refer to the clinical details. This is to avoid an initial bias based on the clinician’s assessment. This concept could also be considered best practice for the pathologist working with AI. The pathologist first makes his/her own opinion and then checks if the AI algorithm produced the same result. If so, this would increase the diagnostic certainty.

The role of AI described above and probably utilizable in the not-to-distant future represents in many ways a variation of the “four-eye principle” often applied in pathology. This refers to at least two pathologists assessing a case before making a diagnosis. In reality, due to time and personnel shortage, the four-eye principle is mainly applied to difficult-to-classify tumors where the resulting treatment and prognosis implications are very high (i.e., cutaneous melanoma). AI could be the “second set of eyes” for both complex and daily routine “simple” diagnoses.

The DigiPath-Viewer software is currently applied in a laboratory on a trial basis. A selection of cases is scanned daily, and our method is evaluated in automatic mode in the background. After the standard daily routine, the dermatopathologists analyze the results of the method and provide feedback, which is used used to further improve the algorithm. While at the moment, only used for research purposes, this kind of tool could transition into clinical routine use.

## 6. Limitations and Future Work

The algorithm could be further fine-tuned by adjusting the thresholds, in particular for distinguishing the two entities SK and BD. However, we believe additional training with higher sample numbers, particularly for difficult-to-classify cases such as irritated SK, would currently be the best approach to improve the algorithm.

A caveat we have repeatedly observed is that the AI makes calls based on the diagnosis it knows. Our algorithm will diagnose SK or BD if it comes across something such as a clear cell acanthoma that it has not been trained to detect. This example demonstrates that, at least for a considerable time to come, pathologists need to remain wary and scrutinize diagnostic calls made by the AI. While misdiagnosing a clear cell acanthoma case is unlikely to have serious consequences, misdiagnosing other entities could have very serious effects for the affected patients.

Optimally, to avoid making false calls and correctly identify anomalous structures, an AI algorithm would likely need to be trained on diverse (preferably all) cutaneous diagnoses and conditions not normally present in the skin (i.e., metastasis from other sites). One diagnosis that we did not include here, although it is related to BD and considerably more prevalent, is actinic keratosis (AK). AKs are early epidermal neoplasias, which can evolve into BD or invasive SCC. An algorithm detecting AK would undoubtedly be of immense help. However, differentiating between normal skin and an initial AK, as well as advanced AK and BD is difficult. These cases in reality often represent a continuum. Therefore, in the current study, we focused on lesions more clearly diverging from one another, making the distinction both for the AI and pathologist relatively clear cut. Having obtained promising results distinguishing SK from DB, we believe increasing the complexity by including AK in the algorithm is the next step we will focus on in the near future. Comparisons of algorithm diagnostic accuracy with levels of agreement between different pathologists are also envisioned. The potential for improvement and work ahead is immense.

## 7. Conclusions

In our study, we assessed the potential of a deep learning approach to distinguish one of the most common benign (SK) from malignant (BD) cutaneous tumors. In addition to classifying sharply demarcated tumors, the established algorithm correctly distinguished different tissue areas of collision tumors on the same histological slide. We evaluated the model’s performance on test sets from three different centers, two of which were not involved in training, and obtained AUC scores > 0.97 in all cases. The model can also distinguish BD and SK from normal tissue. The diagnostic question addressed is focused, and much work remains. However, the algorithm is being continuously improved by adding samples and additional diagnoses such as AK. Despite the potential for further improvement, we believe that the algorithm already represents an aid to the diagnosing pathologist in the current state. Parallel to expanding and improving the algorithm, it is being assessed prospectively on routine cases. We believe this example demonstrates the utility and promise of AI as a routine workhorse tool in dermatopathology.

## Figures and Tables

**Figure 1 cancers-14-03518-f001:**
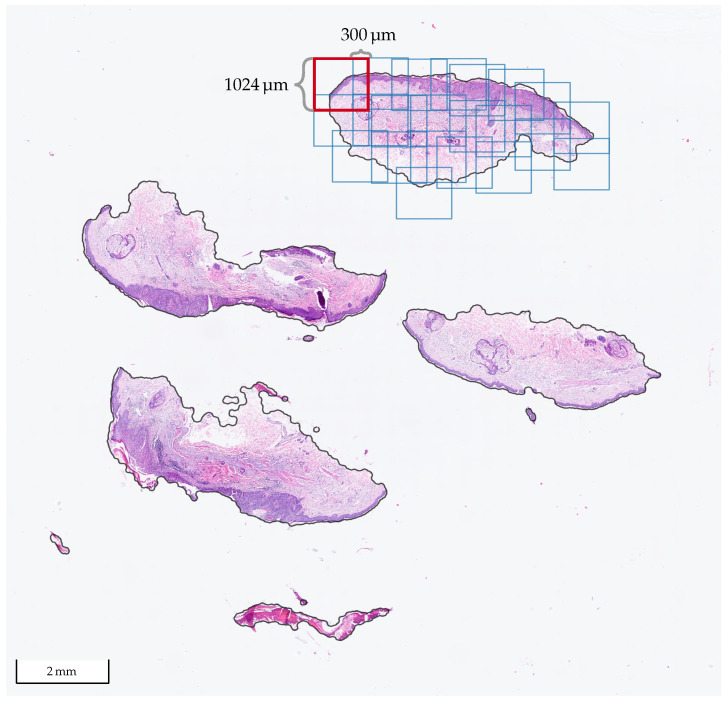
WSI from Center 0 with tissue cuts/sections that contain BD. Prior to the tile extraction, the tissue areas (solid black contour) are detected. The tiles are 1024 μm × 1024 μm at a 10× level of magnification and are extracted with at least 300μm overlap in each direction, as shown in one of the cuts.

**Figure 2 cancers-14-03518-f002:**
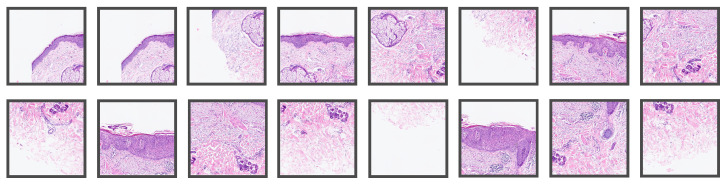
Representative tiles of size 1024 μm × 1024 μm from the WSI in Figure 1 that were used for evaluating the established algorithm.

**Figure 3 cancers-14-03518-f003:**
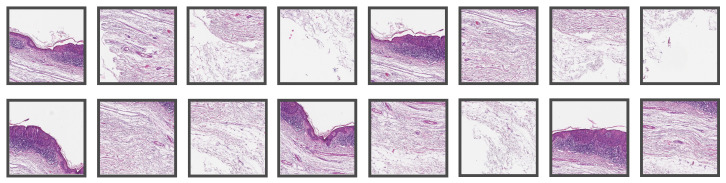
Representative tiles of size 1024 μm × 1024 μm from a WSI from Center 2 that were used for evaluating the established algorithm.

**Figure 4 cancers-14-03518-f004:**
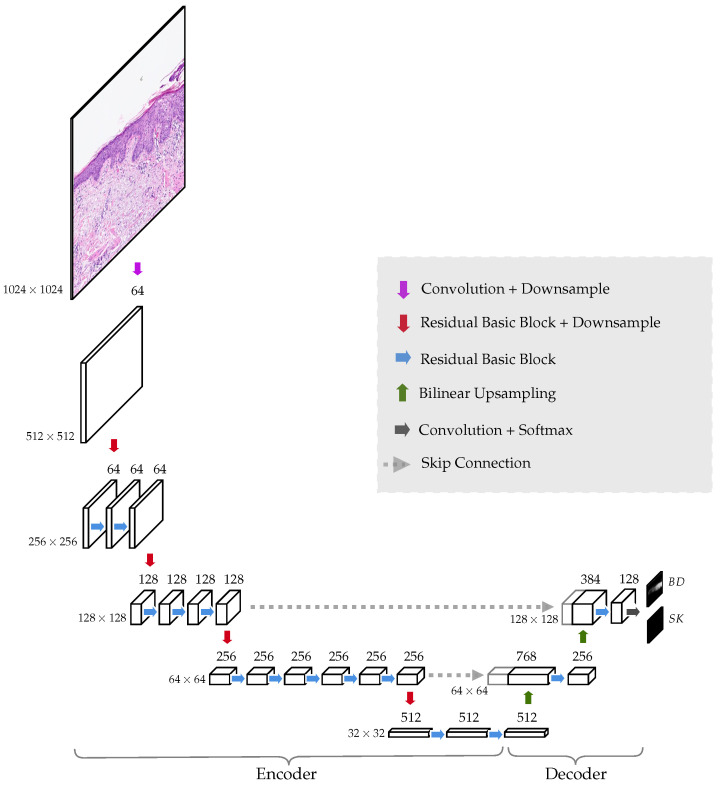
The implemented U-Net-based neural network is composed of an encoder and a decoder. By means of the encoder, the spatial dimension of the WSIs are convoluted and reduced from 1024 px × 1024 px to 32 px × 32 px. This part of the network acts as a feature extractor. Information not relevant for the diagnosis is also discarded due to the bottleneck effect occurring in the middle of the network. The extracted features then pass through a series of convolution and bilinear upsampling operations in the decoder to finally create two probability heat maps of size 128 px × 128 px (one corresponding to BD and the other to SK). Each pixel in the output corresponds to a 8 px × 8 px region of the input image.

**Figure 5 cancers-14-03518-f005:**
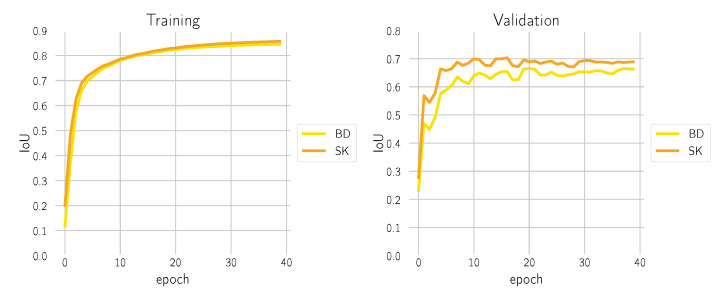
Evaluation of the IoU metric after every training epoch of the model on the training dataset (**left**) and the validation dataset (**right**).

**Figure 6 cancers-14-03518-f006:**
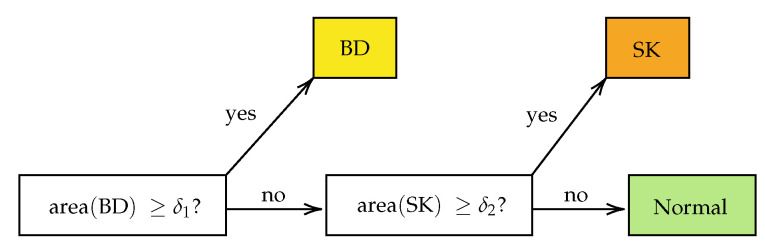
The pipeline for the classification of the tissue cuts is based on the combination of the results from single tiles. If the total area of BD in the tissue section exceeds a threshold $\delta_{1}$, the diagnosis BD is made. If the threshold $\delta_{1}$ is not met, but the area of SK exceeds the threshold $\delta_{2}$, the diagnosis SK is set. Otherwise, the tissue is neither classified as BD nor SK.

**Figure 7 cancers-14-03518-f007:**
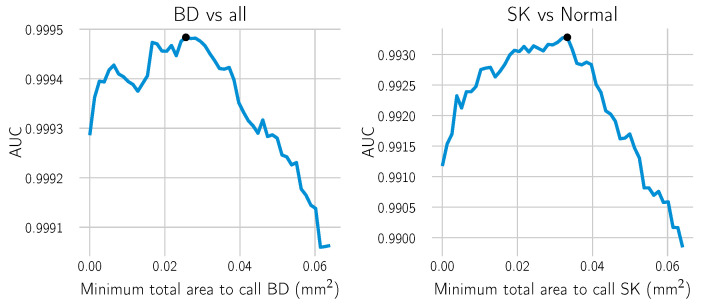
The optimal minimal area to call “BD vs. all” or “SK vs. Normal” was computed using a line search. The AUC was computed for each candidate’s minimal area on the combined training and validation datasets. For each call, the smallest area yielding the highest AUC was selected (black dot).

**Figure 8 cancers-14-03518-f008:**
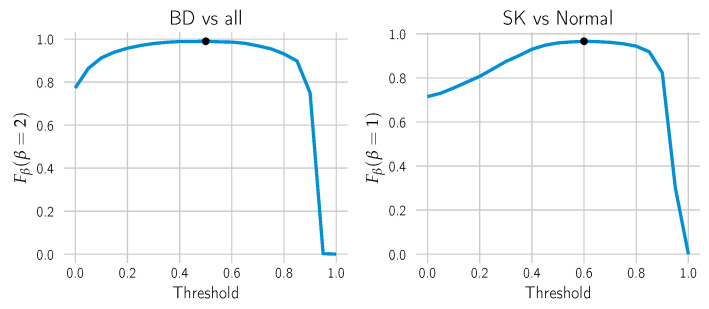
The optimal threshold to call “BD vs. all” or “SK vs. Normal” was computed by line search. The $F_{\beta }$ score was computed for each candidate threshold on the combined training and validation datasets. For each call, the threshold that yields the highest $F_{\beta }$ was selected (black dot).

**Figure 9 cancers-14-03518-f009:**
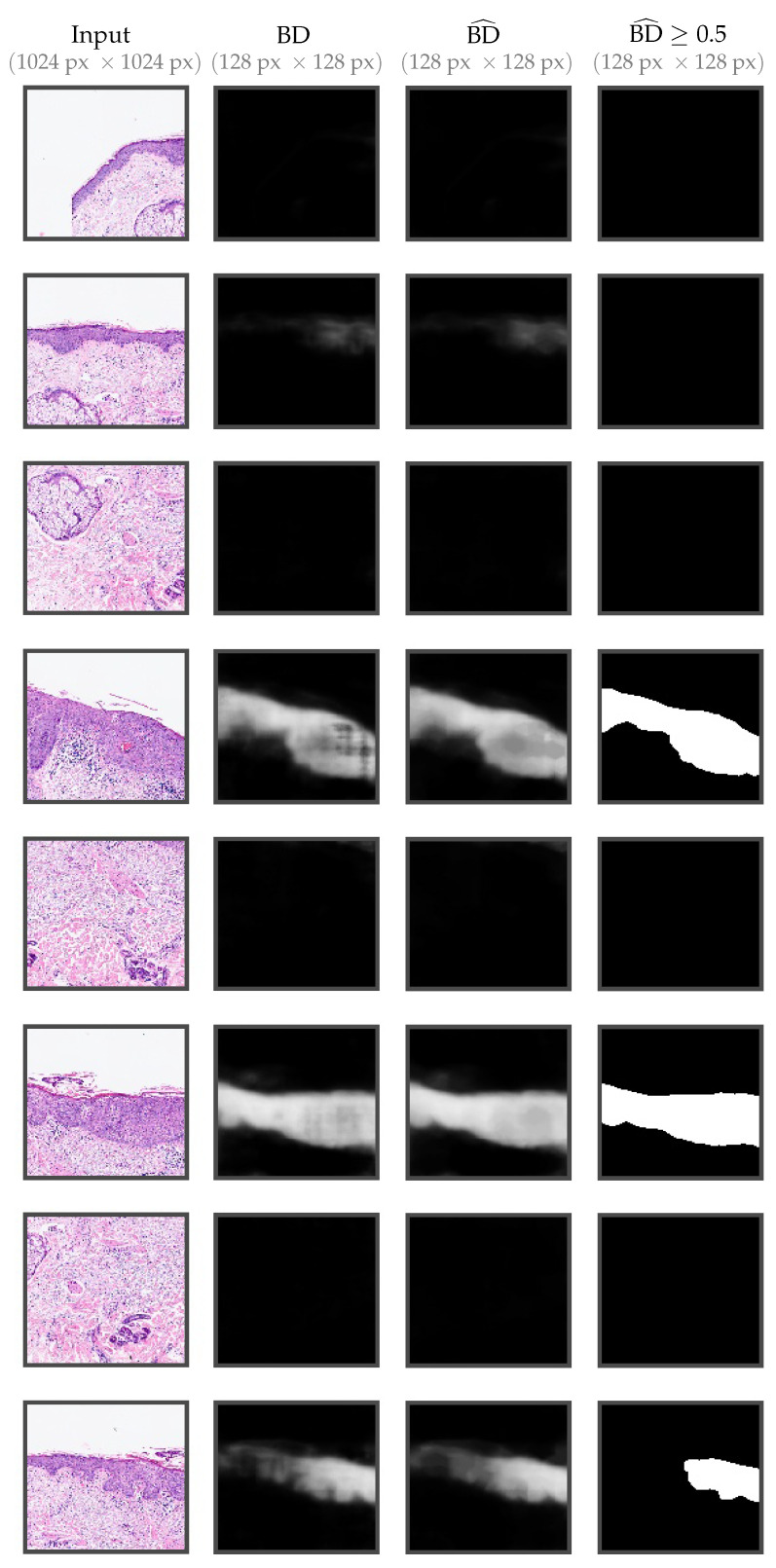
Output examples for some of the tiles ( 1024 μm × 1024 μm) from Figure 2. From left to right: original tile, the output of the network (only BD), post-processing, the binary result after applying a threshold of 0.5.

**Figure 10 cancers-14-03518-f010:**
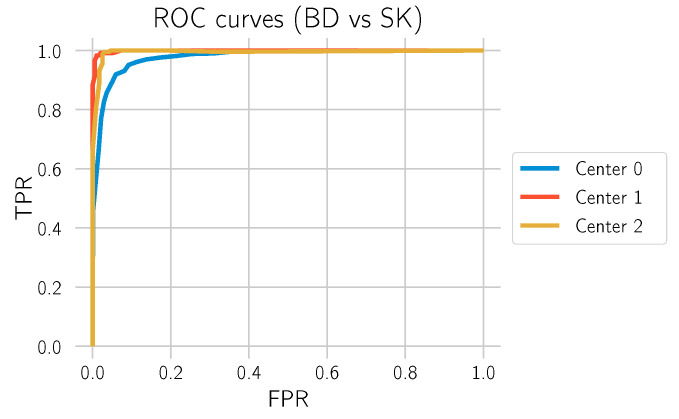
ROC curves for “BD vs. SK” for the 3 different test datasets.

**Figure 11 cancers-14-03518-f011:**
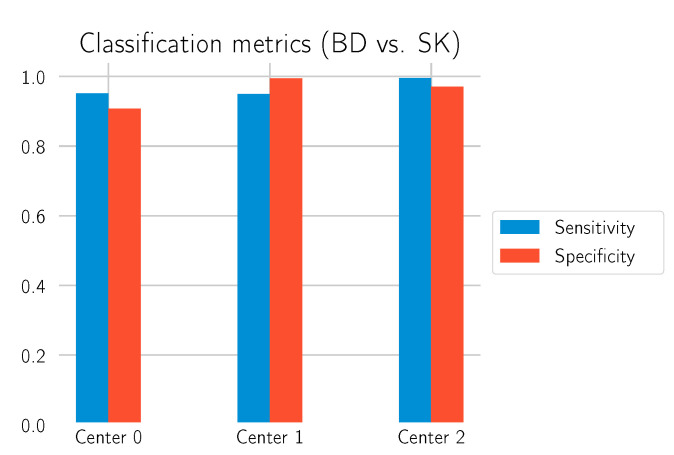
Sensitivity/specificity on the test datasets for the call “BD vs. SK”.

**Figure 12 cancers-14-03518-f012:**
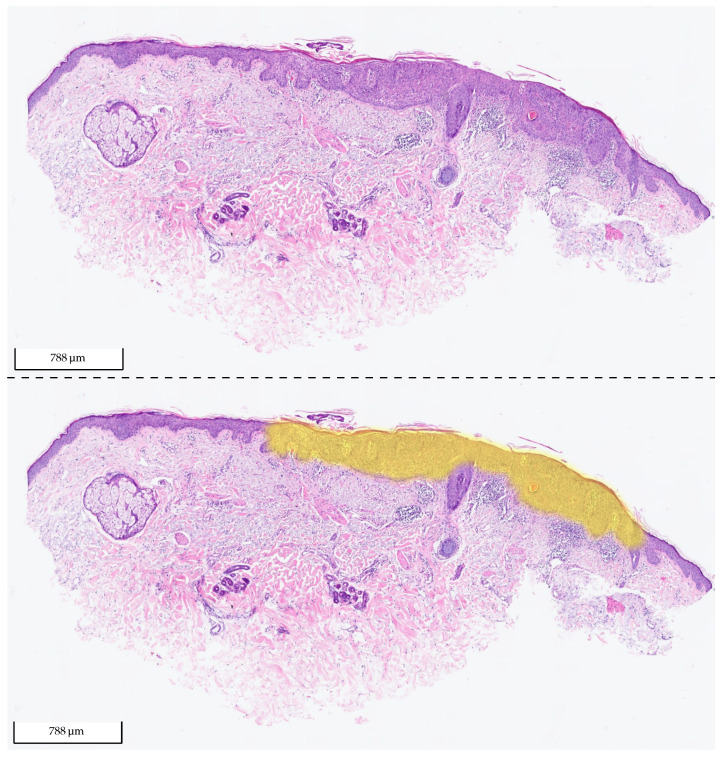
Exemplary tissue section of BD from the test set of Center 0 (**upper picture**) and the same tissue section with its corresponding 2 heat map overlays (**lower picture**): yellow (BD) and orange (SK). The probability value between 0 and 1 is used to set the transparency. As the model did not detect any SK in the section, there is no visible orange (fully transparent). The section was correctly classified by the algorithm as BD.

**Figure 13 cancers-14-03518-f013:**
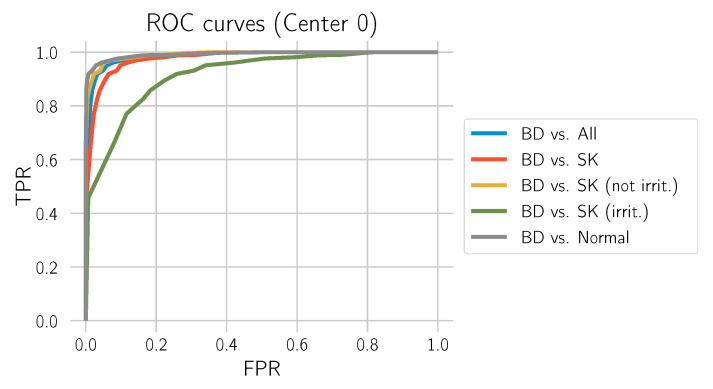
ROC curves for the test set from Center 0.

**Figure 14 cancers-14-03518-f014:**
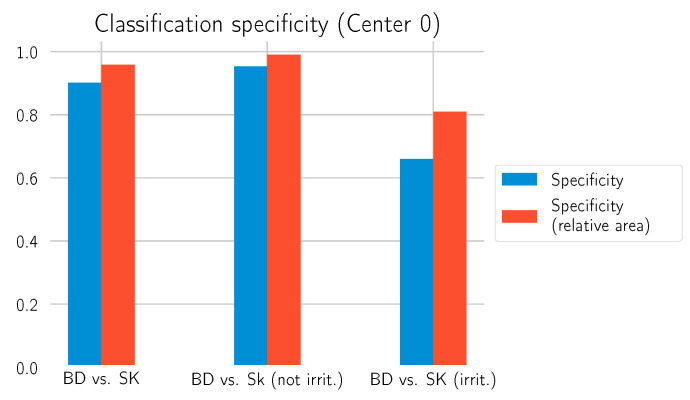
Specificity for the different calls on the test set from Center 0. The plot shows the specificity metric when using the first approach (blue) and the combined approach with the relative area (red).

**Figure 15 cancers-14-03518-f015:**
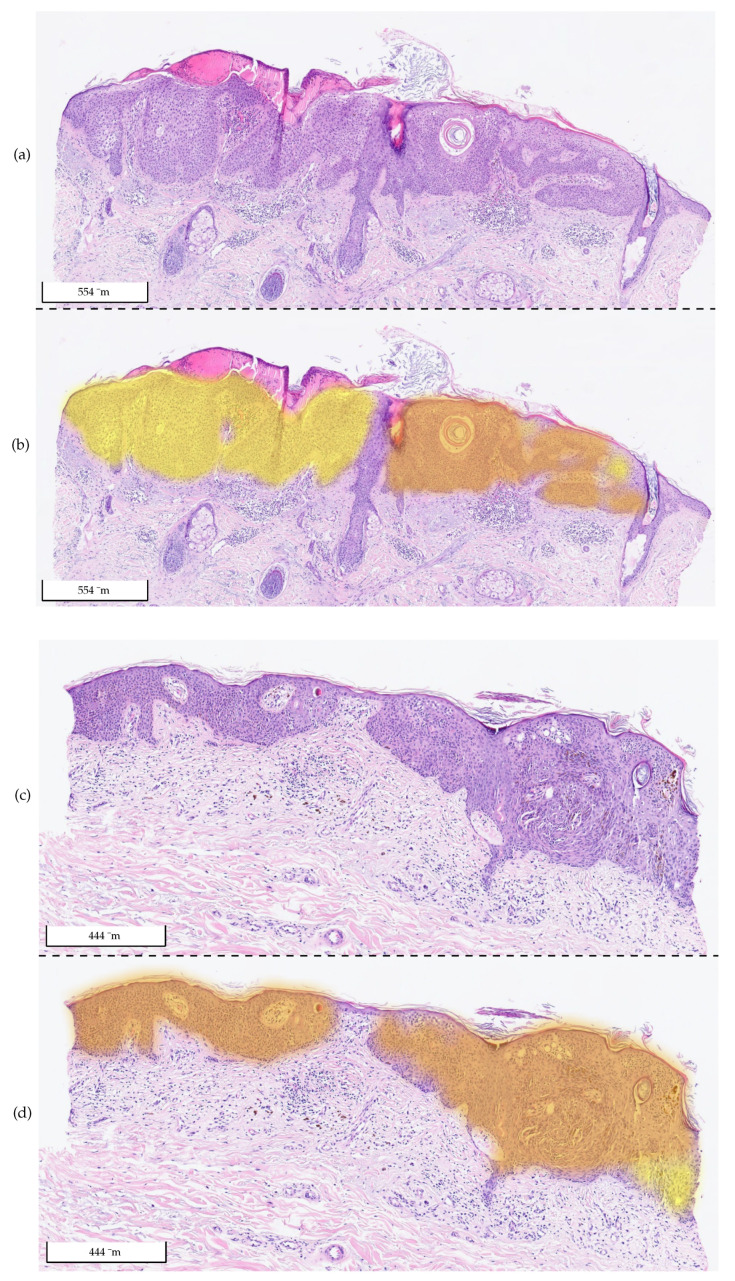
Exemplary tissue sections of two irritated SKs from Center 0 (**a**,**c**) and the same tissue sections with their corresponding 2 overlays (**b**,**d**): yellow (BD) and orange (SK). The probability value between 0 and 1 is used to set the transparency. The relative area for the diagnosis of BD in the first sample (**a**,**b**) is (*r*): 67.7% > 20%, so it was classified as BD, which was incorrect. In the second sample (**c**,**d**), most of the visible color was orange (SK), which was the correct diagnosis. There was a small area (lower right corner) that was identified as BD. The relative area for the diagnosis of BD was (*r*): 5.7% < 20%. Therefore, the section was classified as SK, which was correct.

**Figure 16 cancers-14-03518-f016:**
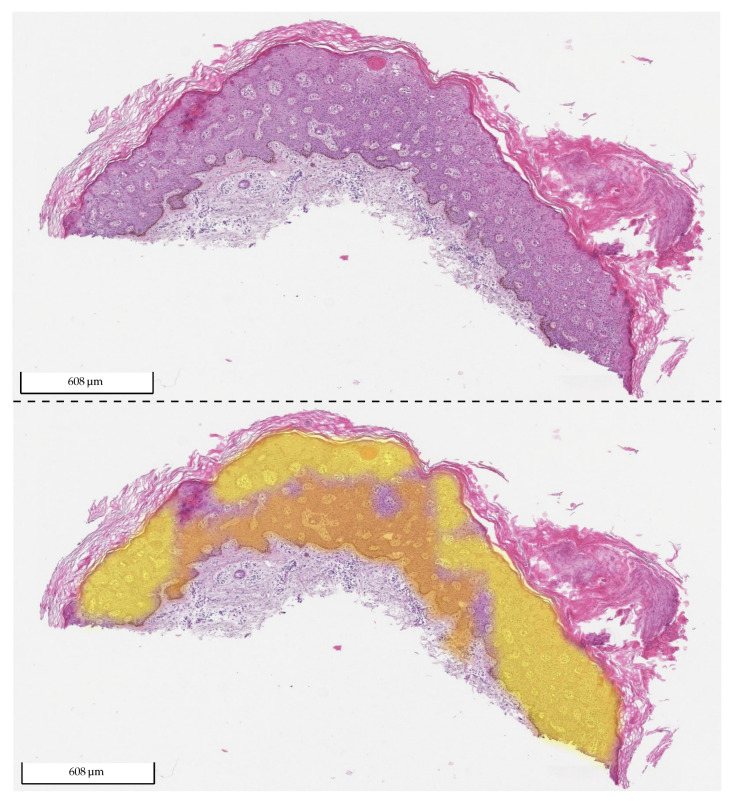
Exemplary tissue section of BD from Center 2 (**upper picture**) and the same tissue section with its corresponding two heat maps overlays (**lower picture**): yellow (BD) and orange (SK). The probability value between 0 and 1 was used to set the transparency. The relative area for the diagnosis of BD was (*r*): 77.04% > 20%. Therefore, it was classified as BD, which was correct.

**Figure 17 cancers-14-03518-f017:**
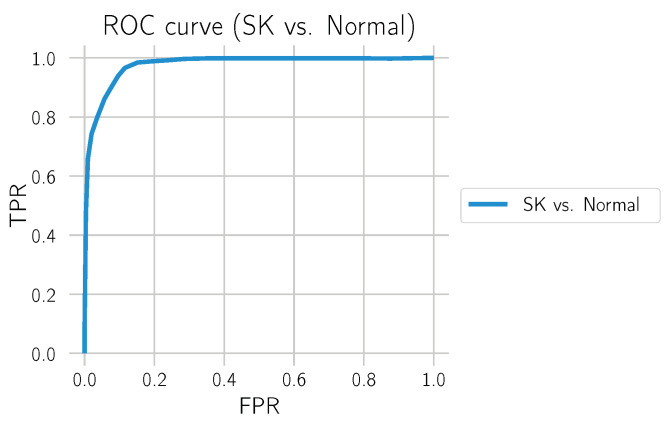
ROC curve for “SK vs. Normal” on the test set from Center 0.

**Figure 18 cancers-14-03518-f018:**
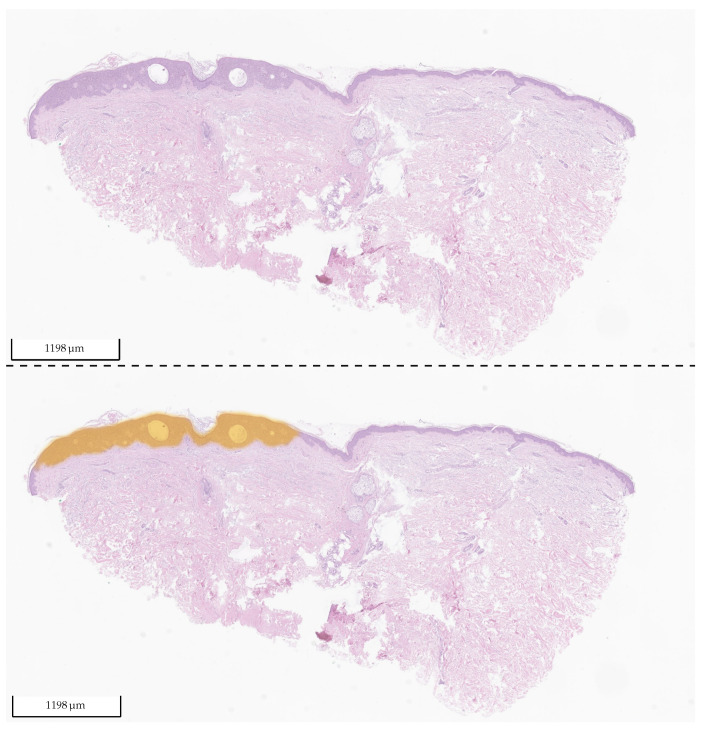
Exemplary tissue section of SK from the test set of Center 1 (**upper picture**) and the same tissue section with its corresponding 2 heat map overlays (**lower picture**): yellow (BD) and orange (SK). The probability value between 0 and 1 is used to set the transparency. As the model did not detect any BD in the section, no yellow is visible (fully transparent). The section was correctly classified by the algorithm as SK.

**Table 1 cancers-14-03518-t001:** Sources of training and test data included in the study. In total, 1935 WSIs were analyzed.

Laboratory	Scanner	Training WSI	Test WSI
Center 0 (Essen)	Hamamatsu Nanozoomer S360	800	865
Center 1 (U. Essen)	Leica^®^ Aperio AT2	-	145
Center 2 (Mainz)	Leica^®^ Aperio AT2	-	125

**Table 2 cancers-14-03518-t002:** Distribution of the data from Center 0. The number of WSIs are provided, as well as the number of tissue sections that were segmented from these, e.g., in Figure 1, there are 4 tissue sections (very small sections are filtered out). Train+Val refers to the part of the data that was used for the training and validation of the model.

Label	Train+Val-WSI	Train+Val-Sections	Test-WSI	Test-Sections
Normal	361	0640	358	0902
BD	215	1086	210	0607
SK	144	0582	254	0691
SK (irrit.)	080	0194	043	0148
Total	800	2502	865	3213

**Table 3 cancers-14-03518-t003:** Selection of the minimal areas and thresholds to call “BD vs. all” and “SK vs. Normal” based on the training dataset.

Call	AUC	Minimum Area	Threshold	Sensitivity	$F_{\beta }$
BD vs. all	0.9994	0.0256 mm^2^ ($\delta_{1}$)	0.5	0.9944	0.9897 (β=2)
SK vs. Normal	0.9933	0.0333 mm^2^ ($\delta_{2}$)	0.6	0.9602	0.9663 (β=1)

**Table 4 cancers-14-03518-t004:** Test scores for “BD vs. SK” for the 3 different test datasets.

Laboratory	AUC	Sensitivity	Specificity
Center 0	0.9774	0.9511	0.9016
Center 1	0.9988	0.9496	0.9948
Center 2	0.9941	0.9957	0.9702

**Table 5 cancers-14-03518-t005:** Test scores for the different calls on the test set from Center 0.

Call	AUC	Sensitivity	Specificity
BD vs. all	0.9858	0.9511	0.9399
BD vs. SK	0.9774	0.9511	0.9016
BD vs. SK (not irrit.)	0.9907	0.9511	0.9534
BD vs. SK (irrit.)	0.9155	0.9511	0.6599
BD vs. Normal	0.9924	0.9511	0.9704

**Table 6 cancers-14-03518-t006:** Test scores for differentiating BD from SK using the relative area approach.

Laboratory	Call	Sensitivity	Specificity
Center 0	BD vs. SK	0.9376	0.9580
Center 0	BD vs. SK (not irrit.)	0.9376	0.9898
Center 0	BD vs. SK (irrit.)	0.9376	0.8095
Center 1	BD vs. SK	0.9076	1.0000
Center 2	BD vs. SK	0.9785	1.0000

**Table 7 cancers-14-03518-t007:** Test scores for “SK vs. Normal” on the test set from Center 0.

Call	AUC	Sensitivity	Specificity
SK vs. Normal	0.9764	0.9394	0.9036

## Data Availability

The data presented in this study are available upon request from the corresponding author.

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
