# Peer review of "Evaluation of a Deep Learning Approach to Differentiate Bowen’s Disease and Seborrheic Keratosis"

_cancers, 2022, doi:10.3390/cancers14143518_

Round 1
Reviewer 1 Report
The paper is overall clearly written.
I only have a few minor comments.
1. The figure captions are sometimes a bit lacking in my opinion. E.g.
- Figs 2,3. Some of the extracted tiles [for training],
- Fig 4. Walk the reader through a little bit, e.g. a U-Net architecture contains an encoder and decoder that function to capture the essence of the image.
- Fig 5. Bit more explanation, e.g. if the area exceed some threshold delta_1 then it is considered a BD
- Fig 6, 7, etc consider removing unseemly grey background
- Table 3 refer to Fig 5
- Fig 10,13 table better
- Fig 11,14,16 etc highlight what colour mean -- e.g. selected by user as ground truth. e.g. Top: without overlay, Bottom: with overlay showing area {predicted by the algorithm/highlighted by the histopathologist). The predicted diagnosis was correct.
2. Literature review. (1.1 Related Methods). Far too short. [18] and [19] state explicitly these were based on (H&E) histopathology (presumably) and not photos, draw attention to more literature (if not present in the most related sense then from a bit further afield). Finally make a statement at the end to the tune of "To the best knowledge of the authors, no work has yet investigated ...."
Author Response
Response to Reviewer 1 Comments
The paper is overall clearly written.
We would like to thank the reviewer for her/his kind assessment of our work. We have adapted the manuscript to overcome the mentioned shortcomings.
I only have a few minor comments.
Point 1: The figure captions are sometimes a bit lacking in my opinion. E.g.
- Figs 2,3. Some of the extracted tiles [for training],
- Fig 4. Walk the reader through a little bit, e.g. a U-Net architecture contains an encoder and decoder that function to capture the essence of the image.
- Fig 5. Bit more explanation, e.g. if the area exceed some threshold delta_1 then it is considered a BD
- Fig 6, 7, etc consider removing unseemly grey background
- Table 3 refer to Fig 5
- Fig 10,13 table better
- Fig 11,14,16 etc highlight what colour mean -- e.g. selected by user as ground truth. e.g. Top: without overlay, Bottom: with overlay showing area {predicted by the algorithm/highlighted by the histopathologist). The predicted diagnosis was correct.
Response 1: Thank you for your comment. We have adapted and extended all legends of all figures and tables including the requested information:
“Figure 2. Representative tiles of size 1024 µm × 1024 µm from the WSI in Figure 1 that were used for evaluating the established algorithm.” (p. 5)
“FIgure 3. Representative tiles of size 1024 µm × 1024 µm from a WSI from Center 2 that were used for evaluating the established algorithm.” (p. 5)
“Figure 4. The implemented U-Net-based neural network is composed of an encoder and a decoder. By means of the encoder the spatial dimension of the WSI are convoluted and reduced from 1024 px × 1024 px to 32 px × 32 px. This part of the network acts as a feature extractor. Information not relevant for the diagnosis is also discarded due to the bottleneck effect occuring in the middle of the network. The extracted features then pass through a series of convolution and bilinear upsampling operations in the decoder to finally create two probability heatmaps of size 128 px × 128 px (one corresponding to BD and the other to SK). Each pixel in the output corresponds to a 8 px × 8 px region of the input image. “ (p. 6)
“Figure 5: Evaluation of the IoU metric after every training epoch of the model on the training dataset (left) and the validation dataset (right).”(p. 7)
[Formerly Figure 5] “Figure 6. The pipeline for the classification of the tissue cuts is based on the combination of the results from single tiles. If the total area of BD in the tissue section exceeds a threshold δ1 the diagnosis BD is made. If the threshold δ1 is not met but the area of SK exceeds the threshold δ2 the diagnosis SK is set. Otherwise the tissue is neither classified as BD nor SK.” (p.8)
The grey background of Figure 7 and Figure 8 [formerly Figure 6 and Figure 7] have been removed. (p.8)
[Formerly Figure 11] “Figure 12. Exemplary tissue section of a BD from the test set of Center 0 (upper picture) and the same tissue section with its corresponding 2 heatmap overlays (lower picture): yellow (BD) and orange (SK). The probability value between 0 and 1 is used to set the transparency. As the model did not detect any SK in the section there is no visible orange (fully transparent). The section was correctly classified by the algorithm as BD.”(p. 11)
[Formerly Figure 14] “Figure 15. Exemplary tissue sections of two irritated SK from Center 0 (a, c) and the same tissue sections with their corresponding 2 heatmap overlays (b, d): yellow (BD) and orange (SK). The probability value between 0 and 1 is used to set the transparency. The relative area for the diagnosis of BD in the first sample (a, b) is (r): 67.7 % > 20 %, so it was classified as BD which was incorrect. In the second sample (c, d), most of the visible color was orange (SK) which was the correct diagnosis. There was a small area (lower right corner) that was identified as BD. The relative area for the diagnosis of BD was (r): 5.7 % < 20 %. Therefore the section was classified as SK which was correct. (p.13)
[Formerly Figure 15] “Figure 16. Exemplary tissue section of a BD from Center 2 (upper picture) and the same tissue section with its corresponding two heatmaps overlays (lower picture): yellow (BD) and orange (SK). The probability value between 0 and 1 was used to set the transparency. The relative area for the diagnosis of BD was (r): 77.04 % > 20 %. Therefore, it was classified as BD which was correct.” (p.14)
Point 2: Literature review. (1.1 Related Methods). Far too short. [18] and [19] state explicitly these were based on (H&E) histopathology (presumably) and not photos, draw attention to more literature (if not present in the most related sense then from a bit further afield). Finally make a statement at the end to the tune of "To the best knowledge of the authors, no work has yet investigated ...."
Response 2: We agree with the reviewer and extended the information on previous project and cited more literature. Additionally, we adapted our conclusion of the tune. It now runs:
“In histopathology, networks based on the U-Net architecture have gained popularity. Numerous working groups obtained impressive results on the CAMELYON challenge [12] for detecting breast cancer metastasis in lymph nodes by using such architectures [13,14]. Li and Lu [15] proposed a model for rapid detection and refined segmentation of breast cancer regions applying a ResNet101 [16] or MobileNetV2 [17] base, followed by a U-Net architecture to refine the segmentation. Recent works have also implemented U-Net variants for the detection and classification of lung cancer [18]. Other, simpler architectures, have also obtained promising results [19,20]. Another architecture called HookNew [21] combines context and details via multiple branches of encoder-decoder convolutional neural networks. In dermatopathology deep learning approaches have been applied for the segmentation of both melanocytic [22] and non-melanoma lesions, incl. basal cell carcinoma (BCC) [23]. A U-Net architecture automatically segmenting the epidermis in histological images has also been implemented [24]. Olsen et al. [20] applied a deep learning-based segmentation method based on the VGG architecture to differentiate three common diagnoses: nodular BCC, SK, and dermal nevus. Ianni et al. [25] trained a model for the differentiation into four groups of lesions: melanocytic, basaloid, squamous, and “other”. Evaluating the model with data from three different laboratories not used for training and obtained a 78 % accuracy was achieved. A two-stage AI-based system for the automatic multi-class grading of oral squamous cell carcinoma and for the segmentation of the epithelial and stromal tissue has also been presented [26]. Despite those numerous studies, significant challenges remain to transfer AI from research into clinical practice [27]. In this work, we present a deep learning solution to differentiate some of the most common diagnoses in dermatopathology: BD and SK. To the best of our knowledge, no work has yet investigated the automatic identification and segmentation of BD, nor its differentiation from SK. These are two of the most common diagnoses seen in the daily clinical routine and their distinction can on occasion be challenging for dermatopathologists. They are both intra-epidermal (in-situ) lesions that can exhibit similar patterns. Our system can assess the WSI and report if it detects any BD or SK lesion at all. This can be very useful to assist the dermatopathologist on the assessment of resection margins. Other works in the literature [20,25,26] mainly focus on analyzing WSIs that contain some lesion and the task is to classify it into 1–4 classes. “ (p.2-3, ll.74 - 105)

Reviewer 2 Report
The paper presents the methodology for the identification of Bowen’s disease and seborrheic keratosis based on deep learning networks. Even though the paper is interesting, it suffers from several stylistic and methodological shortcomings that need to be modified to bring the paper to a level worthy of publication in a journal of this rank.
There are some concerns:
1. In my point opinion, this paper demonstrates a very interesting application of artificial intelligence algorithms in order to differentiate SK from BK. However, in the Introduction, the main contributions of the paper should be outlined better.
2. As the importance of the topic, I recommend updating the ‘Related Methods’, by including the papers recently reported regarding the application of AI in histopathology image analysis (for your convenience: “doi.org/10.1016/j.media.2020.101890”, “doi.org/10.3390/cancers13081784”, “doi.org/10.1002/ctm2.129”). The purpose of this section is to provide an overview of the existing AI–solutions in a similar domain. It is very useful and is part of the introduction/background.
3. For the diagnosis of SK or BD, did you use some kind of metric to determine the degree of agreement between the pathologists (for example, kappa coefficient)?
4. Also, you mentioned that IoU was monitored during the training, both on the training and validation dataset, so I suggest adding those results. From my point of view, it would be interesting if you could broaden the comparison of your results with those obtained by previous studies.
5. Moreover, limitations of the study are not given. A more extensive discussion on this subject would be necessary.
6. Furthermore, I am missing the “Conclusions”. The part with conclusions should be well written, with good sense, and with the final “future perspective” clearly presented.
7. Please add a legend of the abbreviations used.
8. The content of this paper is well structured, English needs revision for minor spelling/grammar errors.
Author Response
Response to Reviewer 2 Comments
The paper presents the methodology for the identification of Bowen’s disease and seborrheic keratosis based on deep learning networks. Even though the paper is interesting, it suffers from several stylistic and methodological shortcomings that need to be modified to bring the paper to a level worthy of publication in a journal of this rank.
We would like to thank the reviewer for her/his kind assessment of our work.
There are some concerns:
Point 1: In my point opinion, this paper demonstrates a very interesting application of artificial intelligence algorithms in order to differentiate SK from BK. However, in the Introduction, the main contributions of the paper should be outlined better
and
Point 2: As the importance of the topic, I recommend updating the ‘Related Methods’, by including the papers recently reported regarding the application of AI in histopathology image analysis (for your convenience: “doi.org/10.1016/j.media.2020.101890”, “doi.org/10.3390/cancers13081784”, “doi.org/10.1002/ctm2.129”). The purpose of this section is to provide an overview of the existing AI–solutions in a similar domain. It is very useful and is part of the introduction/background.
Response 1 / 2: We restructured the introduction and especially added more detailed information on previous work (including the three recommended references) to emphasize the main contributions of our project. It now runs:”“In histopathology, networks based on the U-Net architecture have gained popularity. Numerous working groups obtained impressive results on the CAMELYON challenge [12] for detecting breast cancer metastasis in lymph nodes by using such architectures [13,14]. Li and Lu [15] proposed a model for rapid detection and refined segmentation of breast cancer regions applying a ResNet101 [16] or MobileNetV2 [17] base, followed by a U-Net architecture to refine the segmentation. Recent works have also implemented U-Net variants for the detection and classification of lung cancer [18]. Other, simpler architectures, have also obtained promising results [19,20]. Another architecture called HookNew [21] combines context and details via multiple branches of encoder-decoder convolutional neural networks. In dermatopathology deep learning approaches have been applied for the segmentation of both melanocytic [22] and non-melanoma lesions, incl. basal cell carcinoma (BCC) [23]. A U-Net architecture automatically segmenting the epidermis in histological images 82 has also been implemented [24]. Olsen et al. [20] applied a deep learning-based segmen- 83 tation method based on the VGG architecture to differentiate three common diagnoses: nodular BCC, SK, and dermal nevus. Ianni et al. [25] trained a model for the differentiation into four groups of lesions: melanocytic, basaloid, squamous, and “other”. Evaluating the model with data from 3 different laboratories not used for training and obtained a 78 % accuracy was achieved. A two-stage AI-based system for the automatic multi-class grading of oral squamous cell carcinoma and for the segmentation of the epithelial and stromal tissue has also been presented [26]. Despite those numerous studies, significant challenges remain to transfer AI from research into clinical practice [27]. In this work, we present a deep learning solution to differentiate some of the most common diagnoses in dermatopathology: BD and SK. To the best of our knowledge, no work has yet investigated the automatic identification and segmentation of BD, nor its differentiation from SK. These are two of the most common diagnoses seen in the daily clinical routine and their distinction can on occasion be challenging for dermatopathologists. They are both intra-epidermal (in-situ) lesions that can exhibit similar patterns. Our system can assess the WSI and report if it detects any BD or SK lesion at all. This can be very useful to assist the dermatopathologist on the assessment of resection margins. Other works in the literature [20,25,26] mainly focus on analyzing WSIs that contain some lesion and the task is to classify it into 1–4 classes. “ (p.2-3, ll.74 - 105)
Point 3: For the diagnosis of SK or BD, did you use some kind of metric to determine the degree of agreement between the pathologists (for example, kappa coefficient)?
Response 3: Distinguishing SK from BD is pretty straightforward and not controversial (compared to other diagnoses where diagnostic views between pathologists can vary significantly). We only included cases where a second pathologist verified the initial diagnosis. We did not record cases excluded based on differing pathologist opinions. They were, however, exceedingly rare. We agree performing statistics comparing pathologists consensus, preferably with higher numbers of pathologists evaluating each case, would be interesting in future studies.. We have added this to the limitation section of the study.
It now reads: “Optimally, to avoid making false calls and correctly identify anomalous structures, an 311 AI algorithm would likely need to be trained on diverse (preferably all) cutaneous diagnoses 312 and conditions not normally present in the skin (i.e., metastasis from other sites). One 313 diagnosis which we did not include here, although it is related to BD and considerably more 314 prevalent, is actinic keratosis (AK). AKs are early epidermal neoplasias, which can evolve 315 into BD or invasive SCC. An algorithm detecting AK would undoubtedly be of immense 316 help. However, differentiating between normal skin and an initial AK as well as advanced 317 AK and BD is difficult. These cases in reality often represent a continuum. Therefore, in the 318 current study, we focused on lesions more clearly diverging from one another, making a 319 distinction both for the AI and pathologist relatively clear cut. Having obtained promising 320 results distinguishing SK from DB, we believe increasing the complexity by including 321 AK in the algorithm is the next step we will focus on in the near future. Comparisons of 322 algorithm diagnostic accuracy with levels of agreement between different pathologists are 323 also envisioned. The potential for improvement and work ahead is immense.” (p.17, ll. 311-324)
Point 4: Also, you mentioned that IoU was monitored during the training, both on the training and validation dataset, so I suggest adding those results. From my point of view, it would be interesting if you could broaden the comparison of your results with those obtained by previous studies.
Response 4: Thank you for your feedback. In the related work section we have included more references and mention some differences with our work. We think that a direct comparison in terms of metrics is not possible because to the best of our knowledge there is no other publication that deals with the same problem as we do. We composed an additional figure to demonstrate the IoU metric after every training epoch of the model (p. 7, Figure 5).
Point 5: Moreover, limitations of the study are not given. A more extensive discussion on this subject would be necessary.
Response 5: We fully agree with the reviewer and restructured the manuscript. A new section on the limitations of our study has been included in the manuscript. It now runs:” The algorithm could be further fine-tuned by adjusting the thresholds, in particular for distinguishing the two entities SK and BD. However, we believe additional training with higher sample numbers, particularly for difficult-to-classify cases such as irritated SK, would currently be the best approach to improve the algorithm. A caveat we have repeatedly observed is that AI makes calls based on the diagnosis it knows. Our algorithm will diagnose an SK or BD if it comes across something like a clear cell acanthoma that it has not been trained to detect. This example demonstrates that, at least for a considerable time to come, pathologists need to remain wary and scrutinize diagnostic calls made by AI. While misdiagnosing a clear cell acanthoma case is unlikely to have no serious consequences, misdiagnosing other entities could have very serious effects for affected patients. Optimally, to avoid making false calls and correctly identify anomalous structures, an AI algorithm would likely need to be trained on diverse (preferably all) cutaneous diagnoses and conditions not normally present in the skin (i.e., metastasis from other sites). One diagnosis which we did not include here, although it is related to BD and considerably more prevalent, is actinic keratosis (AK). AKs are early epidermal neoplasias, which can evolve into BD or invasive SCC. An algorithm detecting AK would undoubtedly be of immense help. However, differentiating between normal skin and an initial AK as well as advanced AK and BD is difficult. These cases in reality often represent a continuum. Therefore, in the current study, we focused on lesions more clearly diverging from one another, making a distinction both for the AI and pathologist relatively clear cut. Having obtained promising results distinguishing SK from DB, we believe increasing the complexity by including AK in the algorithm is the next step we will focus on in the near future. Comparisons of algorithm diagnostic accuracy with levels of agreement between different pathologists are also envisioned. The potential for improvement and work ahead is immense.” (p.17, ll. 300-324)
Point 6: Furthermore, I am missing the “Conclusions”. The part with conclusions should be well written, with good sense, and with the final “future perspective” clearly presented
Response 6: In addition to providing more detailed information on limitations that shall be overcome by our future work presented, we restructured and extended our conclusions. We have included the section “Conclusions”: “In our study, we assessed the potential of a deep learning approach to distinguish one 326 of the most common benign (SK) from malignant (BD) cutaneous tumors. In addition to classifying sharply demarcated tumors, the established algorithm correctly distinguished different tissue areas of collision tumors on the same histological slide. We evaluated the model’s performance on test sets from three different centers, two of which were not involved in training, and obtained AUC scores > 0.97 in all cases. The model can also distinguish BD and SK from normal tissue. The diagnostic question addressed is focused, and much work remains. However, the algorithm is being continuously improved by adding samples and additional diagnoses such as AK. Despite the potential for further improvement, we believe that the algorithm already represents an aid to the diagnosing pathologist in the current state. Parallel to expanding and improving the algorithm, it is assessed prospectively on routine cases. We believe this example demonstrates the utility and promise of AI as a routine workhorse tool in Dermatopathology.” (p. 17, ll. 326 - 338)
Point 7: Please add a legend of the abbreviations used.
Response 7: We have included a table depicting the legend of abbreviations at the end of the manuscript. (p.18)
WSI Whole Slide Image
BD Bowen’s Disease
SK Seborrheic Keratosis
AK Actinic keratosis
SCC Squamous Cell Carcinoma
BCC Basal Cell Carcinoma
CNN Convolutional neural network
AI Artificial intelligence
ROC Receiver operating characteristic
AUC Area under the curve
Point 8: The content of this paper is well structured, English needs revision for minor spelling/grammar errors.
Response 8: Thank you for your feedback. We asked a native English speaker to review our manuscript.

Reviewer 3 Report
Dear Editor,
I know that the scope of this article is not within the areas of interest of Cancers journal.
Because the work is not related to cancer by any means. as the section indicates " The development of new methods and technologies for early diagnosis and effective treatment and monitoring of cancer is essential for the improvement of prognosis and overall survival of patients. In this section of Cancers, we welcome contributions from researchers describing new methods and technologies for diagnosis, treatment or monitoring of all types of cancer. Both original articles and reviews are welcome. Topics include, but are not limited to, the following: methods of biomarker detection, proteomics, genomics, transcriptomics, metabolomics, multi-omics, systems biology, imaging technologies, biophotonics, biosensors, nanotechnology, microfluidic technology, bioinformatics, chemometrics, machine learning, artificial intelligence, emerging technologies and new applications of existing technologies."
So, I suggest for the authors to submit it to related journal.
Best Regards
Author Response
Response to Reviewer 3 Comments
Dear Editor,
I know that the scope of this article is not within the areas of interest of Cancers journal.
Because the work is not related to cancer by any means. as the section indicates " The development of new methods and technologies for early diagnosis and effective treatment and monitoring of cancer is essential for the improvement of prognosis and overall survival of patients. In this section of Cancers, we welcome contributions from researchers describing new methods and technologies for diagnosis, treatment or monitoring of all types of cancer. Both original articles and reviews are welcome. Topics include, but are not limited to, the following: methods of biomarker detection, proteomics, genomics, transcriptomics, metabolomics, multi-omics, systems biology, imaging technologies, biophotonics, biosensors, nanotechnology, microfluidic technology, bioinformatics, chemometrics, machine learning, artificial intelligence, emerging technologies and new applications of existing technologies."
So, I suggest for the authors to submit it to related journal.
Response 1: As the reviewer correctly points out, this section of the journal Cancers focuses on new methods for diagnosing cancer. Our entire manuscript focuses on applying artificial intelligence as a novel diagnostic approach to correctly diagnose Bowen´s disease, one of the most frequent cutaneous cancers.
Editor's Response: Bowen’s Disease is one type of skin cancer.
Therefore, this paper is within scope of my special issue.

Round 2
Reviewer 2 Report
In my opinion, this study has a very interesting application for identifying Bowen’s disease and seborrheic keratosis based on deep learning networks. After revision, it is justified for publication in a journal of this rank.